# A Meal with Ultra-Processed Foods Leads to a Faster Rate of Intake and to a Lesser Decrease in the Capacity to Eat When Compared to a Similar, Matched Meal Without Ultra-Processed Foods

**DOI:** 10.3390/nu16244398

**Published:** 2024-12-21

**Authors:** Maria Bárbara Galdino-Silva, Karine Maria Moreira Almeida, Ana Debora Santos de Oliveira, João Victor Laurindo dos Santos, Mateus de Lima Macena, Dafiny Rodrigues Silva, Micnéias Roberth Pereira, André Eduardo Silva-Júnior, Débora Cavalcante Ferro, Déborah Tenório da Costa Paula, Guilherme César de Oliveira Carvalho, Marianna Victória Cerqueira Rocha, Juliane Pereira da Silva, Emiliano de Oliveira Barreto, Nassib Bezerra Bueno

**Affiliations:** 1Laboratório de Nutrição e Metabolismo (LANUM), Faculdade de Nutrição, Universidade Federal de Alagoas, Campus AC Simões, Av. Lourival Melo Mota, s/n, Cidade Universitária, Maceió 57072-900, AL, Brazil; barbaragaldiino@gmail.com (M.B.G.-S.); kariinealmeida@hotmail.com (K.M.M.A.); ana.debora@fanut.ufal.br (A.D.S.d.O.); joao.santos@fanut.ufal.br (J.V.L.d.S.); m.l.macena@hotmail.com (M.d.L.M.); dafiny.rodrigues@outlook.com (D.R.S.); micneias.pereira@fanut.ufal.br (M.R.P.); andreeduardojr@hotmail.com (A.E.S.-J.); debora.ferro@fanut.ufal.br (D.C.F.); deeborah.teenorio@gmail.com (D.T.d.C.P.); guilherme.carvalho@fanut.ufal.br (G.C.d.O.C.); marianna.rocha@eenf.ufal.br (M.V.C.R.); 2Laboratório de Biologia Celular (LBC), Instituto de Ciências Biológicas e da Saúde, Universidade Federal de Alagoas, Campus AC Simões, Av. Lourival Melo Mota, s/n, Cidade Universitária, Maceió 57072-900, AL, Brazil; juliane.silva@icbs.ufal.br (J.P.d.S.); emilianobarreto@icbs.ufal.br (E.d.O.B.)

**Keywords:** food intake, appetite regulation, energy metabolism, heart rate, hormones, insulin resistance

## Abstract

**Background/Objectives**: It is unknown whether the negative health effects associated with ultra-processed foods (UPFs) are due to their nutritional composition or to the extent of food processing itself. We evaluated the impact of a test meal composed only of UPF, according to the NOVA classification, compared to a similar meal without UPF in adults with obesity. **Methods**: This is a parallel, randomized trial. Adult individuals with obesity, according to BMI, % body fat, and/or waist circumference were included. Individuals ate one out of two test meals, matched for energy density, macronutrients, sodium, and fiber, differing in NOVA classification, as a breakfast after a 12-h fast. The rate of intake, appetite, satiety hormones, energy expenditure, and autonomic function were measured. Data were analyzed using mixed analysis of variance. **Results**: Forty-two individuals were included. We found a significantly faster intake rate (07:52 ± 3:00 vs. 11:07 ± 03:16 min), with less chewing and bites, and greater capacity to eat (39.68 ± 22.69 vs. 23.95 ± 18.92 mm) after the UPF meal, without observed differences in the metabolic outcomes. In an exploratory analysis, after adjusting by sex, leptin levels showed a greater decrease after the test meal in the control group. **Conclusions**: Although we found a faster intake rate in the UPF meal, only marginal effects were found on the participants’ capacity to eat after the UPF meal. The high similarity between meals, despite differences according to the NOVA classification, may explain these results. As our study was small, these findings require further investigation.

## 1. Introduction

Ultra-processed foods (UPFs), according to the definition of the NOVA classification, are industrial formulations entirely or mostly composed of ingredients extracted from foods, such as fats and oils, proteins, starches, and sugars, or derived from food constituents (hydrogenated fat, hydrolyzed protein), usually with the addition of other substances (stabilizers, preservatives, flavor enhancers, emulsifiers, among others) not commonly used in culinary preparations. These substances are added with the aim of omitting unwanted sensory qualities while increasing the palatability and attractiveness of the final product [1]. The increase in UPF intake has been identified as a dietary risk factor for several chronic non-communicable diseases, such as obesity [2,3].

Nevertheless, there are still important gaps in knowledge concerning the mechanisms that underlie such association. For example, it is not known whether the worse nutritional composition regarding refined carbohydrates, fats, and salt, usually found in many UPFs, are actually responsible for their undesired health effects or if the extent of the food processing level per se may lead to these detrimental effects [4,5]. Furthermore, other characteristics, such as texture, high energy density, and accelerated intake rate, are portrayed as possible mechanisms, but there are still many controversies regarding the effects of such mechanisms [6]. For example, intake rate has been shown to be higher in meals composed of UPF when compared to non-UPF meals [7]. However, it is not known if these differences persist when comparing UPF meals to non-UPF meals with similar foods and matched for macronutrients and energy density.

Although the faster intake rate of UPF is shown in many studies, there is much heterogeneity in such findings, making this aspect an important target of investigation [8], given that it may trigger several physiological signs related to the perception of post-prandial fullness and satiety, hindering the post-prandial signaling to brain circuits and affecting different metabolic pathways [9,10]. Such pathways may include the secretion of satiety hormones. The structural changes in the UPF due to the levels of processing may facilitate the digestion process, possibly leading to altered hormonal responses and an imbalance in the levels of these hormones [11]. However, the influence of acute UPF intake on post-prandial satiety hormones secretion, as well as on insulin sensitivity, needs to be better investigated compared to matched non-UPF meals [7,12].

Other metabolic pathways are less clearly linked to UPF intake, but some putative mechanisms may underline such an association. Post-prandial energy expenditure (PPEE) may be one of these pathways [13,14]. Due to the lower complexity of the food matrix, UPF may require a shorter metabolization time and, therefore, yield a lower PPEE [12]. Nevertheless, as PPEE is highly dependent on the macronutrient composition of the meal, it is unknown if matched UPF and non-UPF meals would cause a different effect in PPEE. Furthermore, energy metabolism and body weight regulation are directly influenced by the autonomic nervous system (ANS), with evidence that obesity leads to a greater sympathetic tonus [15]. The ANS reacts to different meals, as parasympathetic activation is necessary for the process of food digestion, the well-known state of “rest and digestion”, turning the ANS into an important target when investigating the metabolic impact of meals. ANS function can be measured by heart rate variability (HRV), a psychophysiological model that has been associated with obesity, eating behavior, and the metabolic impact of meals. However, the effect of UPF meals on HRV is unknown [16].

Hence, it is clear that intervention studies are necessary to understand better the mechanisms by which UPF leads to obesity. To date, only two randomized clinical trials have been published. Hall et al. (2019) [7] included 20 individuals in a metabolic chamber with an ad libitum diet either rich in UPF or without UPF, followed for two weeks, in a crossover design. This study revealed that individuals in the UPF group showed a higher energy intake than the non-UPF group, as well as differences in intake rate. Hamano et al. [17] conducted an inpatient, randomized crossover study, including individuals with overweight/obesity who were subjected to the consumption of UPF and non-UPF foods for 7 days. The results showed that the intake rate was significantly faster during UPF consumption compared to non-UPF consumption, both in terms of kcal/min and g/min, leading to increased weight gain in the UPF group. However, there are still a few gaps regarding the acute response of satiety hormones to UPF intake, given that both studies only measured such hormones in the fasted state, as well as in the eating rate of such meals, given such studies offered a wide variety of foods to the individuals; hence, the texture of the provided meals were different. Also, the impacts of UPF intake on the PPEE and the ANS were not investigated. Investigating these outcomes may lead to new findings about the effects of UPF consumption on human health [18].

Therefore, the objective of this study was to evaluate the impact of a meal composed only of foods that are classified as UPFs according to the NOVA classification, compared to a meal with similar foods but that are not classified as UPFs, matched for macronutrients, energy density, fibers, and sodium, on appetitive measures, PPEE, autonomic function, satiety hormones, and insulin sensitivity in individuals with obesity. We primarily hypothesize that the UPF meal would show a faster rate of intake, with a decreased ability to suppress feelings of post-prandial hunger. Our secondary hypothesis is that the UPF meal would lead to a less pronounced increase in gastric inhibitory polypeptide (GIP), glucose, insulin, and leptin levels and a smaller decrease in ghrelin levels while leading to lower PPEE and HRV.

## 2. Materials and Methods

### 2.1. Experimental Design

This was a three-hour randomized, parallel clinical trial with two research arms. Due to the nature of the intervention, it cannot be double-blind. The present study is reported in accordance with CONSORT (Consolidated Standards of Reporting Trials) [19].

### 2.2. Location, Sample and Sampling

The research was carried out at the Nutrition and Metabolism Laboratory, linked to the Faculty of Nutrition at UFAL. Sampling was non-probabilistic for convenience. Recruitment took place in July 2023 through announcements on campus, in the Laboratory, on UFAL Instagram, and the university’s official website. Adult individuals (19–60 years old) of both sexes, with obesity, presenting simultaneously two of the following criteria, were included in this study: I. body mass index between 25 and 40 kg/m^2^; II. waist circumference ≥88 cm for women and ≥102 cm for men; III. body fat percentage ≥35% for women and ≥25% for men, determined by electrical bioimpedance. Individuals using chronic medication (antibiotics, antidiabetics, antiretrovirals, immunosuppressants, antidepressants), individuals with celiac disease or sensitive to gluten, vegetarians or vegans, with restrictions on any food in the test meal, who had any condition that made it impossible to perform the anthropometric test or measurement of energy expenditure components, women in menopause, pregnant or lactating women, or who have undergone any surgical intervention to lose weight were not included. Failure to consume the test meal in full was an exclusion criterion.

### 2.3. Randomization and Allocation

A random sequence of numbers was generated using the Runif function of the statistical software R v.4.3.1 (R Foundation for Statistical Computing, Vienna, Austria). This sequence contained a hundred random numbers varying from 0 to 1. Numbers were rounded to the next integer (0 or 1), and 0 represented an allocation to the control group and 1 to the UPF group. This random sequence of a hundred numbers was kept by a member of the research group who was not involved with the screening, inclusion, or allocation of the participants. Whenever one participant agreed to be included in the study, and all of their baseline data were collected, and on the morning of the test meal, the next random number in the random sequence was revealed by the researcher not involved with inclusion, allocating the participant to either the control or UPF group, to guarantee allocation concealment.

### 2.4. Procedures

The individuals arrived at the Laboratory after fasting for 12 h, abstaining from physical exercise, caffeine, and energy drinks for the last 24 h. The following data were collected: anthropometric and body composition, resting energy expenditure, autonomic function, and appetitive measurements, such as subjective sensations of hunger, satiety, satisfaction, and capacity to eat, in addition to blood collection for biochemical measurement. After the measurements, participants received the test meals. The rate of ingestion and the number of bites and chews were quantified using a stopwatch and a manual counter during the ingestion process. After meals, data on energy expenditure, autonomic function, appetitive measures, and how pleasant and familiar the meal was, as well as blood, were collected again to measure biochemical markers (Figure 1).

On the test day, participants arrived in the lab between 7 a.m. and 9 a.m. Meals (breakfast) took place privately in the Laboratory between 8 a.m. and 10 a.m. so that participants could eat in a silent, peaceful environment with a controlled temperature (22–24 °C). Subjects were instructed to eat the entire meal, and the eating process was recorded for later analysis.

### 2.5. Test Meals

Two test meals were compared: (a) UPF Meal (composed only of UPF, Figure 2a) and (b) Control Meal (consisting of non-UPF, Figure 2b). The meals had similar amounts of kilocalories, macronutrients, fiber, and sodium, differing only in the classification between UPF and non-UPF (Table 1). The meals contained approximately 550 kcal, 70 g of carbohydrates, 22 g of fat, 18 g of protein, 9 g of fiber, and 1300 mg of sodium (Table 1). Depending on the group in which the participant was allocated, they received the UPF meal (containing only UPF), with an energy density of 1.4 kcal/g, or the control meal (containing fresh, minimally processed, processed foods or culinary ingredients), with an energy density of 1.45 kcal/g. The UPF meal consisted of commercial toast with cheese, ham, commercial strawberry jam and margarine, and commercial guava juice with a fiber supplement. The control meal consisted of local bread toast with extra virgin olive oil, fried chicken eggs with soybean oil and salt, and guava fruit juice with honey. Meals were prepared one hour before consumption in the Technical and Dietetics Laboratory of the Faculty of Nutrition/UFAL to comply with health and safety standards when handling food.

### 2.6. Ingestion Rhythm

The amount of time taken to consume the test meal was recorded, from the beginning of the first bite to the last swallow, with the aid of a device with a camera for real-time monitoring purposes. With this, the indices of grams consumed/minute, milliliters consumed/minute, and calories consumed/minute (solid and liquid foods) were calculated using the remainder-intake calculation when necessary. The number of bites and chews was also quantified using a manual counter (statistical counter—WESTERN^®^—CNE-01, Beijing, China) [7,20]. Two independent observers carried out the count, and the average between them was assumed as long as the difference was not greater than 2.5%.

### 2.7. Appetitive Measures

Subjective sensations of hunger, fullness, satisfaction, and capacity to eat were collected through four questions: (1) “How hungry do you feel now?”; (2) “How full do you feel right now?”; (3) “How much do you want to eat now?”; and (4) “How much do you think you can eat now?”. The individuals were asked to answer the questions using a visual analog scale (VAS) ranging from 0 to 100 mm, in which the number 0 corresponded to “nothing/none” and 100 to “excessive” [7,12]. These questions were asked when the participants were in a 12-h fasting, then up to 5 min after the test meal, and 90 min after the test meal. Still using the VAS, participants were asked how pleasant and familiar the test meal was within 5 min of consuming the food.

### 2.8. Energy Expenditure Measurements

Resting energy expenditure (kcal) was measured using an indirect calorimeter (Fitmate RMR, COSMED, Rome, Italy). This procedure occurred after a 12-h fast and immediately before the test meal. The collection site was silent, with dim lighting, and at a comfortable temperature to avoid changes caused by cold or anxiety. On this occasion, measurements of axillary temperature were collected using a clinical digital thermometer (Techline, São Paulo, Brazil) and heart rate using a tensiometer (HEM-4030, OMRON, Tokyo, Japan), in order to avoid measurements with a calorimeter in individuals who showed signs of fever or increased heart rate. The participants put on the calorimeter’s silicone mask, and the volumes of inspired and exhaled oxygen were recorded for 11 min after resting for 10 min in the supine position. The first minute of measurement was discarded, and only the other 10 min were used for analysis to avoid discrepant values due to unfamiliarity with the location and the use of a silicone mask [21]. After measuring oxygen volumes in milliliters per minute, the equation proposed by Weir (1949) [22] was used to obtain energy expenditure at rest for one day: [(3.9 × VO_2_) + (1.1 × VCO_2_)] × 1440. A respiratory coefficient, equivalent to fasting, of 0.85 was considered.

After the measurement of the resting energy expenditure, participants received one out of the two test meals, depending on the allocated group. Then, 75 min after finishing their test meal, the participants were once again subjected to the measurement of oxygen volume using a calorimeter, making it possible to verify PPEE and the thermic effect of food, performing a simple subtraction between PPEE and resting energy expenditure [11]. The post-prandial respiratory coefficient, according to the test meal, was equivalent to 0.87, using the formula: (P × 0.84) + (F × 0.71) + (C × 1.0) [23].

### 2.9. Autonomic Function

Participants underwent HRV analysis using a heart rate monitor (Polar H10, São Paulo, Brazil). This is a non-invasive method that uses an elastic strap in the thoracic region, and the data is transmitted via Bluetooth^®^ to a cell phone with an application (Elite HRV, v. 5.5.8—Elite HRV, Asheville, NC, USA) for recording [24]. Participants remained in the lying position for 5 min to rest and for another 5 min to record. The indices obtained referred to the time domain analysis using the RR: rMSSD—expressed in ms; SD2 and SD1 ratio (SD2/SD1); and analysis in the frequency domain: low- and high-frequency ratio (LF/HF ratio) [25,26]. Collection took place on an empty stomach, immediately before measuring energy expenditure at rest and after the test meal, with measurements between minutes 5 and 15, 35 and 45, as well as 65 and 75 min. The analysis of these data was carried out using the Kubios HRV software (v. Standard 3.5.0, Kuopio, Finland), using its default settings (300 s window and FFT function).

### 2.10. Hormone Analysis

A venous puncture was performed in the cubital fossa of each participant, removing 1 mL of blood, on average, to measure the hormones ghrelin, leptin, and GIP, using the Human Metabolic Hormone Magnetic Bead Panel—Metabolism Multiplex Assay (HMHEMAG—34K—Millipore Corporation, Billerica, MA, USA) [27]. Protease inhibitors for ghrelin (Pefabloc Serine Protease Inhibitor—Sigma, Darmstadt, Germany) were added (1 mg/mL) to the EDTA tubes immediately before collection (Aldrich, St. Louis, MO, USA). In addition, 5 mL of blood was also collected and transferred to a tube with a separating gel to measure glucose and insulin. Such collections occurred at two moments: (1) after fasting for 12 h and (2) 90 min after the test meal by a trained professional. Then, the blood was centrifuged for 10 min at 1000× *g* (80-2B analog centrifuge, 12 tubes—DAIKI, Nanjing China). After centrifugation, the plasma was extracted, stored in microtubes, and stored in a freezer at −80 °C until analysis [28]. The analyses of gastrointestinal hormones were carried out at the Cell Biology Laboratory of the Federal University of Alagoas, according to the manufacturer’s recommendations. Analyses were performed in duplicate, and intra-assay coefficients of variation were calculated. Insulin and glucose measurements were performed in a third-party laboratory. Additionally, the Homeostasis Model Assessment for Insulin Resistance (HOMA-IR) was calculated as recommended by Matthews et al. (1985) [29] [HOMA-RI = fasting insulinemia (mU/L) × fasting blood glucose (mmol/L)/22.5].

### 2.11. Additional Data

Sex, age, education, alcohol consumption, smoking habit, use of chronic medications, presence of chronic diseases, and socioeconomic classification according to the Brazilian Economic Classification Criteria [30] were collected using a questionnaire developed and previously tested by the research group.

In addition to: (1) weight (kg)—measured using a portable, digital scale, with 150 kg and 100 g accuracy, with participants wearing light clothing and barefoot; (2) height (m)—using a portable stadiometer, with a maximum height of 2.2 m and fractions of 1 mm; (3) circumference waist size (cm)—measured with the aid of an inextensible measuring tape, the waist circumference obtained was at the midpoint of the distance between the last rib and the anterior superior iliac crest with the participant in anatomical position; and (4) body composition by electrical bioimpedance (percentage of body fat, mass free of fat and body water)—estimated from tetrapolar bioelectrical impedance RJL Quantum IV (RJL Systems Inc., Clinton, MI, USA).

### 2.12. Sample Size Calculation

Estimating a standardized effect size (Cohen-d) of 0.9, indicating a “large” effect size, considering the post-prandial measure of hunger as the primary outcome, with a statistical power of 80% and a significance level of 5%, 21 patients per group were needed to find significant results.

### 2.13. Statistical Analysis

The data were analyzed using mixed analysis of variance, in which the independent factor was the designated group (meal rich in UPFs vs. meal without UPFs), the dependent factor was the time of measurement (before and after the intervention), and the dependent variables were the outcomes already mentioned. The “*t*” test was also used to compare means between groups, the Levene test to verify the homogeneity of variances, graphical analysis for normality (QQ-plot), and logarithmic transformation of non-normal data. Data are presented as mean and standard deviation for continuous variables and frequency for categorical variables, adopting an alpha value equal to 5%. Furthermore, all analyses were conducted using SPSS software for Windows version 25.0 (IBM, Armonk, NY, USA).

## 3. Results

Forty-two individuals with obesity were included in the study, and females were more predominant at 71.4% (n = 30). No individual was a smoker or had chronic illnesses, nor did they show signs of fever or increased heart rate. Figure 3 presents the flowchart for recruitment of participants in the study. The baseline characteristics of the individuals are presented in Table 2, and there are no statistically significant differences between the groups.

Regarding the intake rates of the meals, there was a statistically significant difference between individuals who consumed a UPF meal compared to the control group (*p* < 0.01). In the UPF group, it was observed that there was a shorter average meal consumption time, fewer bites and chews, and greater consumption of grams, milliliters, and calories per minute, both of solid and liquid foods (Table 3).

There were no significant results regarding appetitive measures at the different moments, except for the variables familiarity (*p* < 0.01) and capacity to eat (*p* = 0.02) (Table 3 and Table 4 and Figure 4).

After logarithmic transformation due to strong non-normality, there was a significant interaction between group and moment in the variable capacity to eat (*p* = 0.04) (Table 4 and Figure 4). In a sensitivity analysis adjusting by sex, this interaction ceases to exist (*p* = 0.12).

Regarding the PPEE measurements, two individuals were not able to repeat such measurements after the meal, so data for 40 individuals were analyzed. Although the mean EE significantly increased after the test meals, there was no statistically significant difference between groups over time (*p* = 0.21; Table 5). Hormonal analysis may also be found in Table 5.

Due to the limited number of the Human Metabolic Hormone Magnetic Bead Panel—Metabolism Multiplex Assay kit available, ghrelin, leptin, and GIP were analyzed in 39 individuals only. Also, one individual in the UPF meal group was unable to measure post-prandial biochemicals, and another individual was excluded from the insulin analysis due to a very discrepant value. Therefore, a total of 41 glucose and 40 insulin samples were analyzed. Due to not being normally distributed, leptin and GIP values were log-transformed, whereas ghrelin values were square-root transformed for analysis. After the test meals, serum ghrelin and leptin levels significantly decreased, while glucose and HOMA-IR increased, but all remained statistically similar between groups.

In an exploratory analysis, there was a significant interaction between the leptin levels over time and the sex of the individuals, in which women showed greater values than men both in the fasting and post-prandial moment (*p*-moment × sex < 0.01), so when adjusting the model for sex, the interaction between moment × group became significant for leptin (*p*-moment × group adjusted by sex = 0.01), with the following back-transformed estimated marginal means: UPF meal fasting = 14.1 [10.6; 18.9] ng/dL; post-prandial = 7.9 [5.9; 10.5] ng/dL; Control Meal fasting = 13.5 [9.5; 19.4] ng/dL; post-prandial = 5.4 [3.8; 7.8] ng/dL, indicating that leptin levels decreased less after the UPF test-meal, when adjusting by sex. To explore this finding further, we split the dataset according to sex. In the women-only analysis, leptin levels went from 21.7 [15.7; 29.9] ng/dL to 13.3 [9.1; 19.4] ng/dL in the UPF group and from 18.1 [13.5.; 24.0] ng/dL to 10.1 [7.2; 14.2] ng/dL, without interactions between group × moment (*p* = 0.34). In the men-only analysis, leptin levels went from 9.2 [4.9; 17.2] ng/dL to 4.7 [2.9; 7.5] ng/dL in the UPF group and from 10.2 [4.2; 24.7] ng/dL to 2.9 [1.5; 5.7] ng/dL (*p*-group × moment = 0.09), indicating that the lower capacity of the UPF meal to decrease leptin would only be seen in men and not in women if our study had enough statistical power. The insulin and GIP levels showed increases in both groups without a statistically significant interaction between groups and time (Table 5). Regarding the HRV analysis, all parameters behaved equally between the groups without significant differences (Table 6).

Regarding the statistical power attained by the present study, calculated from the values for the post-prandial “hunger” measure at 90 min, we found a Cohen-d of 0.50, which is a “medium” effect size, indicating that the study was underpowered to find the “large” effect size computed a priori.

## 4. Discussion

### 4.1. Main Findings

The intake rhythm, appetitive measures, satiety hormones, insulin sensitivity, PPEE, and autonomic function of individuals with obesity were analyzed after a test meal composed only of UPF, compared to a meal without UPF but with similar foods. It was observed that the UPF meal was less able to reduce subjects’ capacity to eat after a meal, in addition to providing for the consumption of greater amounts of grams and calories in a shorter period; however, it was not possible to observe differences in autonomic function, satiety hormones, energy expenditure, and insulin sensitivity markers. In an exploratory analysis, after adjusting by sex, leptin levels showed greater decreases after the control meal. Our main hypothesis, that UPF meals would lead to increased hunger feelings after consumption, was not confirmed. The marked similarities between meals in our study may partially explain our findings.

### 4.2. Methodological Considerations Regarding the Meals

Given the marked absence of metabolic differences between groups after the test meals found in our study, it is important to make some methodological considerations regarding the chosen meals that may partially explain such findings. Firstly, the meals in our study were markedly similar in terms of type of food. We tried to match the meals as much as possible, given that the randomized trials by Hall et al. (2019) [7] and Hamano et al. (2024) [17] did not match the type of food offered to the individuals, and many differences found in their study could be due to the different food matrices offered. In our study, the sources of carbohydrates in each meal (bread and juice in both groups and honey versus jelly) were especially paired. Nevertheless, we understand that the carbohydrate base chosen for the meals (bread from a local bakery vs. industrialized toast) may have been too similar to each other, and the choice of another carbohydrate matrix, such as potatoes or cassava, could have brought different findings. Still, it is noteworthy that choosing a tuber to compose the control meal would induce significant discrepancy in the texture of the meal, which would not be the best option to test whether processing level per se leads to detrimental effects.

Secondly, we followed the NOVA definition when choosing these types of bread; hence, the NOVA definition may not be able to differentiate the types of bread and their metabolic impacts in this context. Although the NOVA classification system is widely used in the literature and is adopted by some national guidelines, this system has known limitations in classifying the degree of food processing, mainly due to the lack of consistency in UPF definitions, and it is widely criticized [31,32]. We acknowledge that other classification systems could also be used, although there is still debate about their uses [33]. Thirdly, in order to match the composition of the meals, dietary fiber needed to be supplemented in the UPF group. Fiber supplements and fibers intrinsic to foods may provide similar metabolic benefits, as the physical presence of fiber in the gastrointestinal tract seems to be an important aspect [34,35]. Nevertheless, the supplemented fiber may contribute to the faster ingestion rate of the UPF meal. However, it is not possible to attribute the large effect found in rate of intake only due to the supplementation of fiber.

### 4.3. Ingestion Rhythm and Appetitive Measures

The UPF-rich meal led to a faster rate of intake, higher consumption of grams and calories per minute, and fewer chews and bites. Furthermore, UPFs contributed to higher scores regarding the capacity to eat, resulting in a lesser decrease in the capacity to eat in the UPF group after ingesting the test meal, but without differences in the other appetitive measures (hunger, fullness, and satisfaction). Forde et al. (2020) [8], in a review involving five studies, with 327 foods and mostly UPF, showed that the average energy intake rate was 69.4 ± 3.1 vs. 35.5 ± 4.4 kcal/min for a UPF-meal and a non-processed meal, respectively, indicating that UPF is usually eaten faster. Nevertheless, the authors highlight the extreme variation in energy intake in foods of the UPF classification. In our study, we used similar foods in both meals, differing only in the NOVA classification, and we were still able to find differences in ingestion rhythm. Hamano et al. (2024) [17] suggest that the high energy density, high sugar content, and reduced fiber content of the UPF may be the main mechanisms that explain the finding of higher UPF intake, thus contributing to increased caloric intake and consequent weight gain. Hall et al. (2019) [7] observed a faster intake rate and greater consumption of calories and grams per minute in the UPF group. Nevertheless, it must be acknowledged that the meals offered by them differed between groups regarding the type of food provided, which may show marked differences in the texture of the foods offered. Therefore, we also chose to match the texture of our meals as much as possible based on previous findings by Teo et al. (2022) [36], who analyzed the impact of different textures of both UPFs and minimally processed foods and observed differences in the rate of intake and energy intake in healthy participants, suggesting that the softness characteristics of UPF could favor its faster rate of intake.

Furthermore, Hall et al. (2019) [7] did not demonstrate significant results regarding appetitive measures, including capacity to eat. Considering that our study found an effect in only one of the four appetitive measures (capacity to eat), it is unclear if this finding alone has any physiological impact on the individuals. A systematic review with meta-analysis [37] indicated that a greater number of chews may be associated with reduced self-reported hunger and increased satiety. However, the physiological mechanisms that explain this relationship are not yet conclusive, although it is conjectured that the increase in the number of chews stimulates the release of gut hormones related to satiety. Although we did not directly analyze the relationship between these factors and found a significant effect only on eating capacity, our results may contribute to the understanding of how the level of food processing influences intake and the feeling of satiety. In short, the mechanisms by which UPFs can promote obesity are diverse and not yet fully defined, resulting from the interaction between exposure to these foods and factors intrinsic to individuals, such as gene expression and the formation of reactive oxygen species [38,39].

It is also worth highlighting that familiarity showed significant differences between the groups in the present study, being higher in the control group, and its relationship with food intake has already been reported in the literature [40]. This is a surprising finding, given that even with lower familiarity scores, the UPF group showed a faster rate of intake and a slightly higher capacity to eat after the test meal. Hence, it is questionable to what extent familiarity would be an important factor in the consumption, possibly indicating that individuals may eat UPFs in an equal manner to non-UPFs, regardless of familiarity.

### 4.4. Post-Prandial Energy Expenditure

The meal containing UPF did not induce a difference in PPEE when compared to the control meal. A previous study of healthy women also found similar results in PPEE after consuming a whole meal versus a processed meal [11]. Considering the high processing involved in UPF production, it was expected that this group would obtain a lower PPEE due to the degree of food refinement [41]. Controversially, the study by Mohr et al. (2020) [14] showed that an acute intervention with a replacement meal (shakes and bars) had a higher PPEE when compared to a breakfast with unprocessed foods. Although isocaloric and matched in macronutrients, the replacement meal had a considerably lower fiber content (1.7 × 19.7 g) and higher amounts of total sugar (50.7 × 27.6 g). Given these findings, it appears that PPEE may not be influenced by the extent of food processing and refinement [42].

### 4.5. Autonomic Function

We did not find any effect of our test meals on the individuals’ HRV. A study carried out with healthy individuals tested three meals with different nutritional content but containing foods similar to UPF. However, they did not use the NOVA classification, and it was observed that, regardless of the content, HRV was not influenced by the meals as well [43]. Some studies have revealed that the composition of the meal influences HRV values. Tentolouris et al. (2003) [44] demonstrated that a diet rich in carbohydrates leads to greater activation of the sympathetic system in lean women and that a diet rich in fats seemed to have no effect. Dikariyanto, Smith, Chowienczyk, Berry, and Hall (2020) [45] found that the consumption of a healthy snack based on almonds resulted in greater parasympathetic regulation, improving the suppression of HRV when compared to another with a hyperpalatable profile (rich in fat, sugar and low in fiber). However, although isocaloric, the distribution of macronutrients in the meals of that study was different. In view of what was observed, it seems that the extent of food processing does not impact the sympathovagal balance of the individuals.

### 4.6. Insulin Sensitivity and Satiety Hormones

Individuals who consumed the UPF meal did not show significant differences in glucose, insulin, HOMA-IR, and satiety hormone concentrations when compared to the meal without UPF. Contrasting our findings, the study of Aberg et al. (2020) [46] observed an improvement in blood glucose measurements in adult diabetic individuals with an average BMI for obesity after consuming whole grains when compared to consuming processed grains, while Dioneda et al. (2020) [12] and Hall et al. (2019) [7] found no differences in glycemic responses between meals with different degrees of processing, while Hall et al., (2019) [7] observed a statistically significant decrease in ghrelin and active GIP in the unprocessed diet group. However, no significant results were observed for total GIP and leptin in the groups.

Our exploratory analysis observed an interaction between leptin levels and group after the adjustment by sex. Studies have already shown that the concentrations of this adipokine are usually higher in females. This finding may be related to excess adipose tissue in women and the interaction of leptin secretion with sexual hormones [47]. In our study, a lower decrease in leptin levels in the UPF group was found after adjustment by sex compared to the control group. Specifically, the lower decrease in leptin after the UPF test meal would only be seen in men but not in women. The role of post-prandial leptin levels is still controversial, as there is much heterogeneity in the literature. In general, studies show either a decrease in or maintenance of leptin levels in individuals with obesity after a meal [48,49]. One recent investigation found decreased leptin levels in individuals with obesity (and also in a subanalysis with females only with obesity) after an oral glucose tolerance test but not after an oral fat tolerance test [50]. The authors speculate that leptin might have a post-prandial regulatory role, possibly in fat oxidation in skeletal muscle, that is disrupted in individuals with obesity [50]. In our study, men had higher values for BMI and waist circumference than women, which may indicate that their obesity status was worse and the UPF test meal was not able to decrease their leptin levels. Nevertheless, further studies are warranted.

### 4.7. Limitations

There are some limitations in our study: (1) PPEE was measured over a limited amount of time (90 min), which decreased the ability to find significant differences between groups. Nevertheless, it is known that PPEE peaks between 60 and 180 min, and we were able to capture a significant increase in PPEE over time, as shown by our statistical analysis. Although our study evaluated the effects of PPEE and hormones at two specific moments, pre- and post-intervention, which already allowed us to identify metabolic responses, we understand that results using curves with multiple collections would be the ideal analyses. (2) When calculating our sample size, we anticipated less variability in our results, as well as greater differences between groups. Still, our findings did not allow us to confirm these aspects. Thus, our study ended up being underpowered in finding effects in the outcomes. Nevertheless, we were able to find highly significant effects in the rate of intake measures, indicating that our sample was able to confer some statistical power. Our data on metabolic parameters is useful to guide future research on this theme. (3) We recognize that our study results are based on an acute effect observed after a single meal. Therefore, we suggest caution when interpreting the data, as this specific assessment limits the generalization of the findings. Unlike studies that investigate the effects of chronic UPF consumption, ours does not allow us to capture the potential long-term impacts of this consumption and its repercussions on health. Furthermore, the exclusive inclusion of individuals with obesity may represent an additional limiting factor when generalizing the results to other populations. (4) Due to the nature of the intervention, it would be difficult to blind the individuals to which meal they were going to have (i.e., individuals would probably notice the differences in processing levels, even with similar meals). Hence, we opted not to make any blinding of the participants, making an open-label study that is prone to biases. Finally, one last limitation worth noting is the impossibility of providing financial support to the participants, which is not permitted by the Brazilian Ethics in Research Committee. Making volunteers undergo the data collection process and make long hours of their time available for longer measurements is sensitive. Therefore, carrying out a crossover design, which would expose individuals twice to the experimental protocol, and measuring the PPEE for up to 5 h, as recommended, were discarded options in our study.

## 5. Conclusions

The present study found that a meal composed of UPF according to the NOVA classification leads to a higher rate of intake, fewer chews, and bites, and is also less likely to reduce the capacity to eat of individuals with obesity when compared to a similar control meal, without UPF and matched for macronutrients and energy density. However, this investigation was unable to determine whether the extent of processing leads to harmful effects on the PPEE, autonomic function, satiety hormones, and markers of insulin sensitivity in these individuals. As an exploratory finding, after adjusting the main analysis by sex, a lower decrease in leptin levels after the UPF meal was found, which warrants further investigation. As our study was underpowered, our effect sizes may be considered hypothesis-generating for future studies. Such studies should also focus on specific variations in amounts and types of sugars, fatty acids, and soluble and insoluble fibers, as well as food texture.

## Figures and Tables

**Figure 1 nutrients-16-04398-f001:**
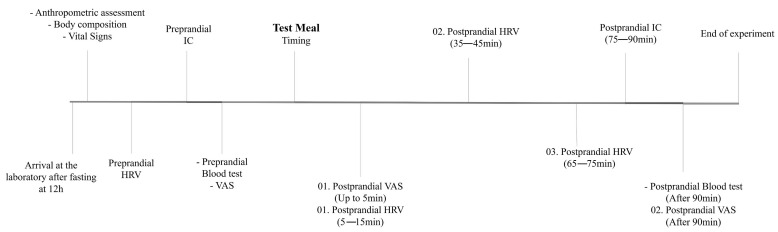
Timeline of procedures performed with participants on the day of the clinical trial.

**Figure 2 nutrients-16-04398-f002:**
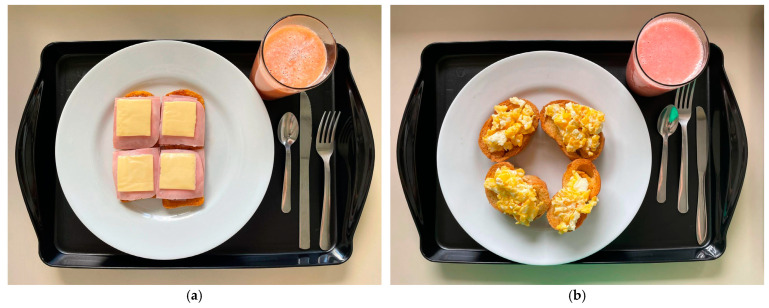
(**a**) Test meal composed of ultra-processed foods offered to the group of participants allocated to the UPF Meal group on the day of the clinical trial. (**b**) Test meal without ultra-processed foods offered to the group of participants allocated to the Control Meal group on the day of the clinical trial.

**Figure 3 nutrients-16-04398-f003:**
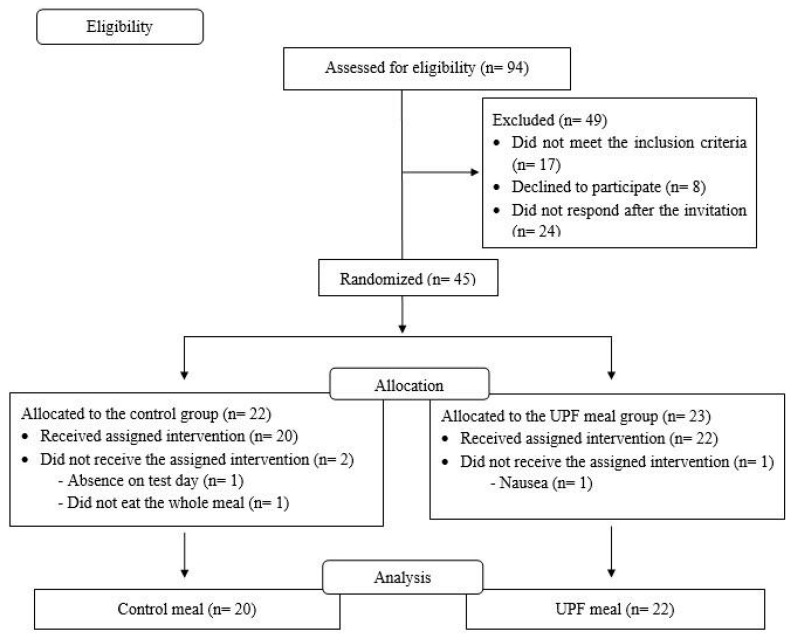
Flowchart of recruitment, randomization, allocation, and analysis of eligible participants for the clinical trial on the effect of meals rich in ultra-processed foods on metabolic parameters in obese individuals.

**Figure 4 nutrients-16-04398-f004:**
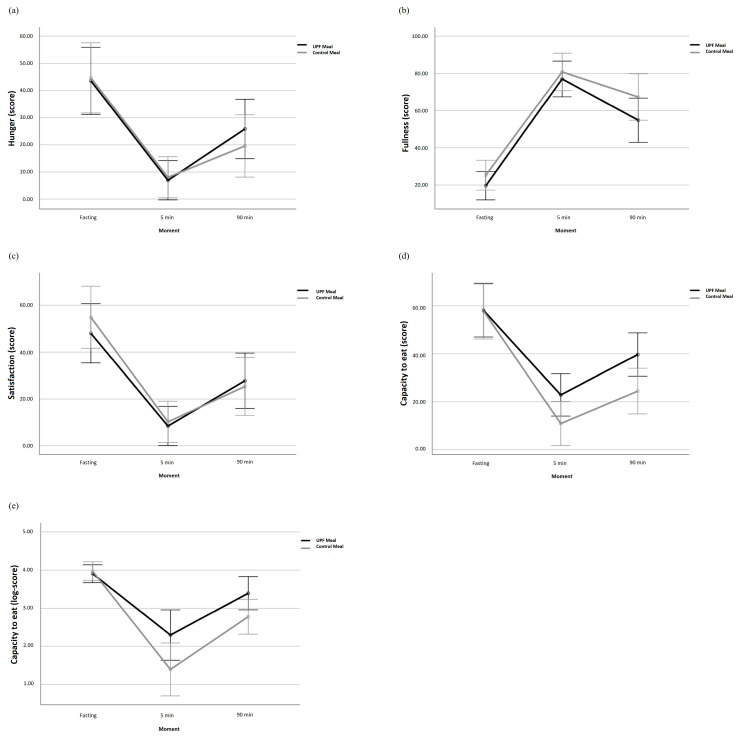
Graphical representation of the scores of appetitive measures after analysis of variance (ANOVA) (**a**): hunger; (**b**): Fullness; (**c**): Satisfaction; (**d**): Capacity to eat; (**e**): Capacity to eat score after logarithmic transformation, before and after meals rich in ultra-processed foods versus meals without ultra-processed foods in individuals with obesity. Data presented as mean and 95%CI. UPF: ultra-processed food.

**Table 1 nutrients-16-04398-t001:** Composition of the test-meals offered to the individuals.

UPF Meal								Control Meal						
Bauducco^®^ Traditional Toast (Guarulhos, SP, Brazil) (04 units)—30 g								Regular bakery bread toast (01 unit)—50 g						
Polenghi^®^ dish cheese (Angatuba, SP, Brazil) (03 fts)—54 g								Chicken egg (02 units)—100 g						
Perdigão^®^ Ham (Marau, RS, Brazil) (2.5 fts)—50 g								Soybean vegetable oil (01 teaspoon)—5 mL						
Purifruti^®^ Strawberry Jelly (Itupeva, SP, Brazil) (02 tablespoons)—40 g								Refined salt (01 coffeespoon)—2 g						
Primor^®^ Margarine (Ipojuca, PE, Brazil) (01 dessertspoon)—15 g								Extra virgin olive oil (01 tablespoon)—8 mL						
Maratá^®^ guava juice (Estância, SE, Brazil) (01 glass)—200 mL								Guava pulp juice (1 glass)—200 mL						
Nestlé^®^ FiberMais fiber module (Araras, SP, Brazil) (1 tablespoon)—9 g								Bee honey (01 tablespoon)—20 g						
Meal	KCAL	ED (kcal/g)	PTN (g)	PTN (%)	CHO (g)	CHO (%)	LIP (g)	LIP (%)	Fiber (g)	Na (mg)	UPF (%)	P (%)	CI (%)	INMP (%)
UPF	557	1.40	18.4	13.2	71.3	51.2	21.9	35.4	8.5	1493	100.0	0.0	0.0	0.0
No UPF	559	1.45	18.7	13.4	70.6	50.5	22.1	35.5	8.5	1267	0.0	26.8	31.5	41.7

UPF: ultra-processed foods; g: grams; fts: slices; mL: milliliters; KCAL: kilocalories; ED: energy density; PTN: proteins; CHO: carbohydrates; LIP: lipids; Na: sodium; P: processed; CI: culinary ingredients; INMP: in natura or minimally processed; mg: milligrams; %: percentage.

**Table 2 nutrients-16-04398-t002:** Descriptive characteristics of the sample.

Variable	Groups				*p*-Value ^a^
	UPF Meal (n = 22)		Control Meal (n = 20)		
	n	%	n	%	
Women	14	63.6	16	80	0.31
Race/skin color					0.65
White	7	31.8	5	25	
Black	1	4.5	3	15	
Brown	14	63.6	12	60	
Alcoholic	10	45.5	9	45	0.99
Education					0.75
Full medium	14	63.6	14	70	
Graduated	8	36.4	6	30	
Economic class					0.47
A/B1/B2	8	36.4	6	30	
C1/C2	11	50	14	70	
DE	3	13.6	0	0	
	Mean	SD	Mean	SD	
Age (years)	28.6	6.71	29.9	7.91	0.55
Height (m)	1.68	0.09	1.64	0.11	0.15
Weight (kg)	89.9	16.36	82.1	15.42	0.12
BMI (kg/m^2^)	31.6	3.82	30.4	3.44	0.30
CC (cm)	97.2	10.24	92.9	12.87	0.23
Fat mass (%)	39.7	6.11	41.4	6.46	0.37

n: sample number; %: percentage; UPF: ultra-processed foods; BMI: body mass index; CC: waist circumference; SD: standard deviation; m: meters; cm: centimeters; kg: kilogram. ^a^ *p*-value for the unpaired “*t*” test; *p* < 0.05 was considered significant.

**Table 3 nutrients-16-04398-t003:** Differences in the rate of intake in test meals.

	UPF Meal (n = 22)		Control Meal (n = 20)		
	$\bar{x}$	SD	$\bar{x}$	SD	*p*-Value ^a^
Consumption time (min:s)	07:52	3:00	11:07	03:16	<0.01
Grams consumed/min	29.98	14.82	18.27	14.74	<0.01
Milliliters consumed/min	30.08	5.56	19.58	6.01	<0.01
Calories consumed/min (solids)	70.47	34.84	46.83	14.25	<0.01
Calories consumed/min (liquids)	13.77	6.83	8.39	2.58	<0.01
Number of bites	27.32	9.68	42.65	11.54	<0.01
Number of chews	424.07	148.50	587.80	152.89	<0.01
Pleasure (mm)	67.18	27.07	74.85	35.38	0.43
Familiarity (mm)	38.18	34.07	80.30	30.90	<0.01

UPF: ultra-processed foods; $\bar{x}$: average; SD: standard deviation; min: minutes; s: seconds; mm: millimeters. ^a^
*p*-value for the unpaired “*t*” test.

**Table 4 nutrients-16-04398-t004:** Appetite measurements before and after the test meal.

	Fast			5 min			90 min			
	UPF Meal	Control Meal		UPF Meal	Control Meal		UPF Meal	Control Meal		
	$\bar{x}$ ± SD	$\bar{x}$ ± SD	*p*-Value ^a^	$\bar{x}$ ± SD	$\bar{x}$ ± SD	*p*-Value ^a^	$\bar{x}$ ± SD	$\bar{x}$ ± SD	*p*-Value ^a^	ANOVA ^b^
Hunger (mm)	43.55 ± 30.34	44.45 ± 26.47	0.91	6.95 ± 8.82	2.85 ± 5.25	0.07	25.86 ± 25.24	15.00 ± 17.16	0.10	0.30
Fullness (mm)	19.55 ± 17.02	25.25 ± 18.64	0.30	76.77 ± 17.51	80.50 ± 26.36	0.58	54.82 ± 25.71	67.20 ± 29.53	0.15	0.59
Satisfaction (mm)	48.00 ± 30.57	54.85 ± 27.86	0.45	8.23 ± 14.41	10.25 ± 23.14	0.73	27.77 ± 24.69	25.35 ± 30.00	0.77	0.64
Capacity to eat (mm)	57.86 ± 28.18	57.80 ± 22.60	0.99	22.86 ± 24.68	10.95 ± 14.23	0.06	39.68 ± 22.69	23.95 ± 18.92	0.02	0.09
Capacity to eat (log)	3.91 ± 0.61	3.97 ± 0.45	0.69	2.29 ± 1.54	1.39 ± 1.53	0.06	3.39 ± 0.94	2.77 ± 1.08	0.05	0.04

$\bar{x}$: average; SD: standard deviation; UPF: ultra-processed foods; mm: millimeters. UPF Meal (n = 22) and Control Meal (n = 20). ^a^ *p*-value for the unpaired “*t*” test; ^b^ *p*-value of mixed ANOVA.

**Table 5 nutrients-16-04398-t005:** Fasting and post-prandial energy expenditure, serum glucose, insulin, HOMA-IR, and satiety hormones levels of participants.

		Fasting		Post-Prandial		*p*-Value	
		Mean	CI 95%	Mean	CI 95%	Time ^1^	Time × Group ^2^
EE	Control	1657	1437; 1877	1906	1671; 2140	<0.01	0.21
(kcal)	UPF	1882	1662; 2101	2207	1973; 2442		
Glucose	Control	78.6	75.1; 82.2	84.6	79.2; 90.1	0.03	0.26
(mg/dL)	UPF	81.0	77.5; 84.5	82.9 ^a^	77.4; 88.4		
Insulin	Control	9.15	7.54; 10.77	29.68	20.76; 38.60	<0.01	0.57
(uUI/mL)	UPF	8.37	6.75; 9.99	32.13 ^b^	23.21; 41.05		
HOMA-IR	Control	0.98	0.81; 1.16	3.13	2.21; 4.05	<0.01	0.57
	UPF	0.91	0.73; 1.08	3.38	2.46; 4.30		
Ghrelin (pg/mL) ^3^	Control	84.6	59.2; 112.3	9.0	5.7; 12.9	<0.01	0.67
	UPF	81.1	57.9; 108.5	10.3	6.9; 14.5		
Leptin (pg/mL) ^3^	Control	16.1	11.5; 22.1	7.8	5.3; 11.3	<0.01	0.21
	UPF	15.4	11.2; 21.1	8.8	6.2; 12.6		
GIP (pg/mL) ^3^	Control	48.9	35.5; 66.6	287.1	223.6; 368.7	<0.01	0.86
	UPF	57.9	42.5; 78.2	330.2	259.8; 424.1		

UPF: ultra-processed foods; kcal: kilocalories; EE: energy expenditure; mg/dL: milligrams/deciliter; uUI/mL: international microunit/milliliter; pg/mL: picograms/mililiter. UPF Meal (n = 22) and Control Meal (n = 20). ^a^ Glucose (n = 21); ^b^ Insulin (n = 20). *p* < 0.05 was considered significant. ^1^
*p*-value for an unadjusted simple mixed-ANCOVA, without groups in the model. ^2^
*p*-value for the interaction between time × group, in a mixed-ANCOVA model. ^3^ Ghrelin data were analyzed using a square root transformation. Leptin and GIP data were analyzed using a natural logarithmic transformation. Data in the table is presented back-transformed.

**Table 6 nutrients-16-04398-t006:** Heart rate variability of the sample.

	Fasting			5–15 min			35–45 min			65–75 min			
	UPF Meal	Control Meal	*p*-Value ^a^	UPF Meal	Control Meal	*p*-Value ^a^	UPF Meal	Control Meal	*p*-Value ^a^	UPF Meal	Control Meal	*p*-Value ^a^	ANOVA^b^
	$\bar{x}$ ± SD	$\bar{x}$ ± SD		$\bar{x}$ ± SD	$\bar{x}$ ± SD		$\bar{x}$ ± SD	$\bar{x}$ ± SD		$\bar{x}$ ± SD	$\bar{x}$ ± SD		
rMSSD (ms)	55.32 ± 33.90	43.06 ± 27.74	0.21	50.74 ± 32.89	35.47 ± 22.29	0.08	43.94 ± 24.60	40.89 ± 29.56	0.71	51.61 ± 40.13	38.21 ± 22.84	0.19	0.29
LF/HF	0.95 ± 0.91	1.48 ± 1.50	0.17	2.06 ± 1.97	2.20 ± 2.06	0.83	1.63 ± 1.34	2.17 ± 2.44	0.37	1.73 ± 1.47	1.96 ± 1.62	0.63	0.78
SD2/SD1	1.57 ± 0.51	1.95 ± 0.74	0.05	1.76 ± 0.57	2.17 ± 0.78	0.06	1.81 ± 0.64	2.20 ± 0.88	0.10	1.75 ± 0.64	1.99 ± 0.68	0.23	0.55

UPF: ultra-processed foods; min: minutes; $\bar{x}$: average; SD: standard deviation; ms: milliseconds; rMSSD: square root of the mean square of the differences between adjacent normal RR intervals; LF: Low Frequency*;* HF: High Frequency; SD1: standard deviation of instantaneous beat-to-beat variability; SD2: long-term standard deviation of continuous RR intervals. UPF Meal (n = 22) and Control Meal (n = 20). ^a^
*p*-value for the unpaired “*t*” test; ^b^
*p*-value for a mixed ANOVA. *p* < 0.05 was considered significant.

## Data Availability

The raw data supporting the conclusions of this article will be made available by the authors on request.

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
