# Peer review of "A Meal with Ultra-Processed Foods Leads to a Faster Rate of Intake and to a Lesser Decrease in the Capacity to Eat When Compared to a Similar, Matched Meal Without Ultra-Processed Foods"

_nutrients, 2024, doi:10.3390/nu16244398_

Round 1
Reviewer 1 Report
Comments and Suggestions for Authors
Thank you for the opportunity to review this manuscript. The study is interesting and informative. The methodology was thorough and informative, although it lacked sufficient power to see differences in many of the variables studied.
Title
I recommend changing the title to state the observed differences between the meals. With insufficient power, you cannot determine the differences between many of the variables examined.
Abstract
The abstract was well-written, but I have the following suggestions:
Line 32 – I recommend re-phrasing the sentence to say that the rate of intake was faster. Calling the ingestion rate “lower” is confusing.
Line 34 – You are in danger of committing a Type 2 error here. I don’t think that you had enough power to see a difference. It would be better to state it as “without observed differences.”
Introduction
Line 61 – Please replace “by” with “of.”
Line 95 – Please replace “is” with “are.”
Line 102 – Please replace “by” with “of.”
Materials and methods
Line 113 – Please replace “is” with “was.”
The procedures look appropriate and are clearly described. The picture of the two meals was informative.
Section 2.8
It is not clear when the resting energy expenditure test was done. Was it conducted immediately before the test meal? Please clarify.
The paragraph that begins on Line 227 is confusing. It says that PPEE was measured after the test meal, but the next sentence mentions how the subjects received one of the two test meals. Then it mentions having the PPEE test 75 to 90 minutes after the test meal. Please clarify.
Section 2.12
Line 282- I am surprised that you thought that the post-prandial measure of hunger with have a “large” effect size. Do you have a reference for this decision? The eating rate might have a large effect size as Hall et al was able to see this difference with a small number of subjects.
Results
Table 2- You two groups appear to be well-matched for descriptive characteristics.
Your descriptions on lines 315-319 are clear and informative. You carefully explained that no differences could be seen between the groups except for familiarity and capacity to eat (lines 324-326) but you need to expand this paragraph by mentioning that familiarity was surprisingly higher in the control group not the UPF group and that capacity to eat was higher in the UPF group.
Lines 370-378- I think that the sex interaction if very interesting. Could you please expand your discussion of this a bit? Do you think the fact that women have more body fat allows for greater variation after eating the meal and therefore increases the likelihood that you could see a difference in leptin? Please forgive me if I misinterpreted your findings and please tell me more so that I can better understand your findings.
Discussion
The discussion was well-written and interesting.
Line 391- Please replace “by” with “of.”
Line 394-395- Be careful here. With insufficient power, you don’t know if the variables responded to these meals differently. Perhaps you just “couldn’t see” any differences (Type 2 error).
Lines 395-396- Please help me understand what you are saying here. Are you saying that when you separated the men’s and women’s values that you saw a difference in leptin in both genders? I do not understand how to interpret this information.
Line 476- “Submitted” is not the right word to use here. I suggest stating it as “Individuals who consumed the UPF meal.”
Line 486 – Please interpret “Our analysis observed an interaction between leptin and group after adjustment by sex.” What does this mean? Perhaps if you provided data in your results section, it would make more sense when you mention it in your discussion section.
Line 490- By “lighter” do you mean “smaller?” “Lighter” is not the right word to use here.
Conclusions
Line 521-524- It would be better to say that “this study was not able to determine if the extent of processing leads to harmful effects . . .” rather than state that it “does not support the fact that. . “
Here is the danger of a Type 2 error.
Thank you for the opportunity to review your paper. It is interesting and informative and adds to the scientific findings regarding UPF. It is not surprising that you could not see significant differences in many of your variables. If UPF had big effects on metabolism it would be easier to show that they are “causing” obesity. I am impressed that you were able to enroll as many subjects as you did without the ability to compensate them.

Author Response
REVIEWER 1:
We would like to appreciate your kind and thoughtful suggestions. Please find our answers below in a point-by-point manner. The comments by the reviewer are in bold. Our response is preceded by a "R-".
Thank you for the opportunity to review this manuscript. The study is interesting and informative. The methodology was thorough and informative, although it lacked sufficient power to see differences in many of the variables studied.
TITLE
I recommend changing the title to state the observed differences between the meals. With insufficient power, you cannot determine the differences between many of the variables examined.
R- We changed the title as suggested. “A meal with ultra-processed foods leads to a faster rate of intake and to a lesser decrease in the capacity to eat when compared to a similar, matched meal without ultra-processed foods.”
ABSTRACT
The abstract was well-written, but I have the following suggestions:
Line 32 – I recommend re-phrasing the sentence to say that the rate of intake was faster. Calling the ingestion rate “lower” is confusing.
R - Thank you for your comment. Modification made as suggested (line 32)
Line 34 – You are in danger of committing a Type 2 error here. I don’t think that you had enough power to see a difference. It would be better to state it as “without observed differences.”
R - Thank you for your comment. Modification made as suggested (line 34)
INTRODUCTION
Line 61 – Please replace “by” with “of.”
R - Thank you for noticing the typo. Modification made as suggested (line 61)
Line 95 – Please replace “is” with “are.”
R - Thank you for noticing the typo. Modification made as suggested (line 98).
Line 102 – Please replace “by” with “of.”
R - Thank you for noticing the typo. Modification made as suggested (line 105).
MATERIALS AND METHODS
Line 113 – Please replace “is” with “was.”
R - Thank you for noticing the typo. Modification made as suggested (line 116).
The procedures look appropriate and are clearly described. The picture of the two meals was informative.
R - Thank you for your comment.
Section 2.8
It is not clear when the resting energy expenditure test was done. Was it conducted immediately before the test meal? Please clarify.
The paragraph that begins on Line 227 is confusing. It says that PPEE was measured after the test meal, but the next sentence mentions how the subjects received one of the two test meals. Then it mentions having the PPEE test 75 to 90 minutes after the test meal. Please clarify.
R- Thank you for your consideration in section 2.8. Some information was inserted (lines 214-215, "and immediately before the test meal" and line 228 “After the measurement of the resting energy expenditure” and lines 229-230 “Then, 75 minutes after finishing their test meal”). Also, some other sentences were excluded ("PPEE was measured after the test meal") and the paragraph was reorganized to facilitate understanding.
Section 2.12
Line 282- I am surprised that you thought that the post-prandial measure of hunger with have a “large” effect size. Do you have a reference for this decision? The eating rate might have a large effect size as Hall et al was able to see this difference with a small number of subjects.
R- We planned (a priori, as may be seen in our clinical trials registry) a standardized “large” effect with 42 subjects in total without specifying any outcomes. Although we did not specifically pre-register “hunger” as the main outcome, we believed that, based on Hall's findings that ad libitum intake of UPF led to overeating (with a large effect size, cohen-d), hunger would also be greater in the UPF group for meals offered with fixed energy. In addition, this large effect was also chosen because we had financial limitations, so a conservative estimate of the effect size would reduce the number of outcomes investigated and, consequently, the scope of our investigation. Nevertheless, considering the instrument used to assess hunger (scale), and the similarity of our interventions, we do believe now that our choice of “hunger” as the primary outcome was inadequate and that our study would only be able to detect differences in rate of intake, indeed. We recognize that this was an inconsistency in our planning.
RESULTS
Table 2- You two groups appear to be well-matched for descriptive characteristics.
Your descriptions on lines 315-319 are clear and informative. You carefully explained that no differences could be seen between the groups except for familiarity and capacity to eat (lines 324-326) but you need to expand this paragraph by mentioning that familiarity was surprisingly higher in the control group not the UPF group and that capacity to eat was higher in the UPF group.
R- Thank you for your consideration. In the original submission, we had mentioned this issue in the discussion section. As we do agree that this is an important issue, we have rephrased this part of the discussion to better highlight this information (lines 493-498, “This is a surprising finding, given that even with lower familiarity scores, the UPF group showed a faster rate of intake and a slightly higher capacity to eat after the test meal. Hence, it is questionable to what extent familiarity would be an important factor in the consumption, possibly indicating that individuals may eat UPF in an equal manner to non-UPF, regardless of familiarity.” ).
Lines 370-378- I think that the sex interaction if very interesting. Could you please expand your discussion of this a bit? Do you think the fact that women have more body fat allows for greater variation after eating the meal and therefore increases the likelihood that you could see a difference in leptin? Please forgive me if I misinterpreted your findings and please tell me more so that I can better understand your findings.
R- We also believe that this finding is intriguing and somewhat difficult to interpret. We have added further information on this finding in the results section (lines 384-391): “To explore this finding further, we split the dataset according to sex. In the women-only analysis, leptin levels went from 21.7 [15.7; 29.9] ng/dL to 13.3 [9.1; 19.4] ng/dL in the UPF group and from 18.1 [13.5.; 24.0] ng/dL to 10.1 [7.2; 14.2] ng/dL, without interactions between group*moment (p = 0.34). In the men-only analysis, leptin levels went from 9.2 [4.9; 17.2] ng/dL to 4.7 [2.9; 7.5] ng/dL in the UPF group and from 10.2 [4.2; 24.7] ng/dL to 2.9 [1.5; 5.7] ng/dL (p-group*moment = 0.09), indicating that the lower capacity of the UPF meal to decrease leptin would only be seen in men and not in women if our study had enough statistical power.”. We also added more on this in our discussion section, lines 540-553 “In our study, a lower decrease in leptin levels in the UPF group was found after adjustment by sex compared to the control group. Specifically, the lower decrease in leptin after the UPF test meal would only be seen in men but not in women. The role of post-prandial leptin levels is still controversial, as there is much heterogeneity in the literature. In general, studies show either a decrease or maintenance of leptin levels in individuals with obesity after a meal [48,49]. One recent investigation found decreased leptin levels in individuals with obesity (and also in a subanalysis with females only with obesity) after an oral glucose tolerance test but not after an oral fat tolerance test [50]. The authors speculate that leptin might have a post-prandial regulatory role, possibly in fat oxidation in skeletal muscle, that is disrupted in individuals with obesity [50]. In our study, men had higher values for BMI and waist circumference than women (data not shown), which may indicate that their obesity status was worse and the UPF test meal was not able to decrease their leptin levels. Nevertheless, further studies are warranted.”
DISCUSSION
The discussion was well-written and interesting.
Line 391- Please replace “by” with “of.”
R - Thank you for noticing the typo. Modification made as suggested (line 415).
Line 394-395- Be careful here. With insufficient power, you don’t know if the variables responded to these meals differently. Perhaps you just “couldn’t see” any differences (Type 2 error).
R - Thank you for your comment. Modification made as suggested (lines 418-419).
Lines 395-396- Please help me understand what you are saying here. Are you saying that when you separated the men’s and women’s values that you saw a difference in leptin in both genders? I do not understand how to interpret this information.
R- We acknowledge that the information was confusing. And the finding itself is also confusing. We tried our best to better describe and discuss what we found (as reported in the commentary above). We hope it is now suitable. Please note that lines 395-396 are still results and not the discussion section. We improved the discussion section (lines 540-553).
Line 476- “Submitted” is not the right word to use here. I suggest stating it as “Individuals who consumed the UPF meal.”
R- Thank you for your comment. Modification made as suggested (line 527).
Line 486 – Please interpret “Our analysis observed an interaction between leptin and group after adjustment by sex.” What does this mean? Perhaps if you provided data in your results section, it would make more sense when you mention it in your discussion section.
R - We appreciate your attention to this matter. We tried to better explain our findings with additional data in the results section (lines 384-391) and discussion section (lines 540-553) as explained in the point above. We hope it is now suitable.
Line 490- By “lighter” do you mean “smaller?” “Lighter” is not the right word to use here.
R- Thank you for your comment. Modification made as suggested (line 541 “lower”).
CONCLUSION
Line 521-524- It would be better to say that “this study was not able to determine if the extent of processing leads to harmful effects . . .” rather than state that it “does not support the fact that. . “Here is the danger of a Type 2 error.
R- Thank you for your comment. Modification made as suggested (line 590).
Thank you for the opportunity to review your paper. It is interesting and informative and adds to the scientific findings regarding UPF. It is not surprising that you could not see significant differences in many of your variables. If UPF had big effects on metabolism it would be easier to show that they are “causing” obesity. I am impressed that you were able to enroll as many subjects as you did without the ability to compensate them.
R- We appreciate your thoughtful and kind suggestions for improving our paper.
Reviewer 2 Report
Comments and Suggestions for Authors
This manuscript presents a randomized clinical trial comparing the acute effects of ultra-processed food (UPF) meals versus non-UPF meals on various metabolic and appetitive outcomes in individuals with obesity. The study addresses an important topic in nutrition science and provides some interesting findings. However, there are several areas that require improvement or clarification.
Major comments
1. The sample size calculation is based on a large effect size (Cohen's d = 0.9) for the primary outcome of post-prandial hunger. This assumption may be overly optimistic, potentially leading to an underpowered study. A more conservative effect size estimate would have been prudent. Future studies should consider using effect sizes from similar published studies or conducting pilot studies to inform sample size calculations. Otherwise, consider conducting a follow-up study with a crossover design to reduce inter-individual variability and increase a statistical power.
2. While the authors attempted to match the meals for macronutrients and energy density, the classification of foods as UPF or non-UPF using the NOVA system may oversimplify the complexity of food processing. Recent literature has highlighted limitations of the NOVA classification system and suggested more nuanced approaches to categorizing processed foods (PMID: 35314769, PMID: 30820487, PMID: 28703086). The authors should discuss these limitations and consider using additional classification systems in future studies.
3. The study focuses on acute effects of a single meal, which may not capture the long-term impacts of UPF consumption on health outcomes. The authors should acknowledge this limitation and discuss how their findings relate to studies examining chronic UPF intake, such as the work by Hall et al. (PMID: 31269427).
4. While the study measured various outcomes, it lacks a clear mechanistic framework to explain the observed differences between UPF and non-UPF meals. The authors should provide more in-depth discussion of potential mechanisms underlying their findings (PMID: 38294671, PMID: 38212644, PMID: 38418082), particularly regarding the faster intake rate and differences in appetitive measures as follows.
-Include additional measures of food reward and hedonic responses to provide a more comprehensive understanding of the appetitive effects of UPF vs. non-UPF meals.
-Explore the potential role of food texture and oral processing in mediating the observed differences in intake rate and appetitive responses.
-Conduct subgroup analyses to investigate potential sex differences in responses to UPF and non-UPF meals, as previous studies have suggested differential effects.
-Consider incorporating measures of gut hormones and microbiome composition in future studies to elucidate potential mechanisms underlying the metabolic effects of UPF consumption.
5. The study focuses on individuals with obesity, which may limit generalizability to other populations. Including a wider range of BMIs could provide more comprehensive insights.
Minor comments
6. More detailed information on the specific foods used in each meal would be helpful for readers to better understand the intervention.
7. The manuscript mentions that meals were consumed between 8 and 10 am. Considering potential diurnal variations in metabolism, standardizing the meal time more strictly could improve consistency.
8. The exploratory analysis of leptin levels adjusted for sex is interesting but should be interpreted with caution due to the post-hoc nature of the analysis.
9. Provide a more detailed rationale for the choice of outcomes measured and their relevance to understanding the health impacts of UPF.
10. The manuscript would benefit from a more detailed description of the randomization process and allocation concealment. In addition, due to the nature of the intervention, the study could not be double-blinded. This potential source of bias should be more thoroughly addressed in the limitations section.
11. The authors should provide more information on the validity and reliability of the methods used for measuring energy expenditure and autonomic function in this population.
12 The discussion section could be strengthened by comparing the findings more extensively with existing literature on UPF consumption and its effects on appetite and metabolism.
13. The authors should address potential confounding factors, such as habitual diet, physical activity levels, and sleep patterns, which could influence the measured outcomes.
14. Provide a more detailed rationale for the choice of outcomes measured and their relevance to understanding the health impacts of UPF.
15. Consider including a table or figure summarizing the main findings to improve the clarity of result presentation.
16. Discuss the implications of the findings for public health and dietary recommendations, while acknowledging the limitations of the acute, single-meal design.
17. Expand the introduction or discussion section by citing the following relevant articles (PMID: 36517001, PMID: 32674529).
Author Response
Thank you for your kind and thoughtful suggestions on our paper. Please find our response to each point raised below:
This manuscript presents a randomized clinical trial comparing the acute effects of ultra-processed food (UPF) meals versus non-UPF meals on various metabolic and appetitive outcomes in individuals with obesity. The study addresses an important topic in nutrition science and provides some interesting findings. However, there are several areas that require improvement or clarification.
R - We appreciate your important suggestions and time spent thoroughly reviewing our paper. We hope that the paper is now suitable for publication.
Major comments
- The sample size calculation is based on a large effect size (Cohen's d = 0.9) for the primary outcome of post-prandial hunger. This assumption may be overly optimistic, potentially leading to an underpowered study. A more conservative effect size estimate would have been prudent. Future studies should consider using effect sizes from similar published studies or conducting pilot studies to inform sample size calculations. Otherwise, consider conducting a follow-up study with a crossover design to reduce inter-individual variability and increase a statistical power.
R- Thank you for the comment. We agree that our study is small, and we have acknowledged it in the abstract and the main text (conclusion). As explained to the other reviewer, We planned (a priori, as may be seen in our clinical trials registry) a standardized “large” effect with 42 subjects in total without specifying any outcomes. Although we did not specifically pre-register “hunger” as the main outcome, we believed that, based on Hall's findings that ad libitum intake of UPF led to overeating (with a large effect size, cohen-d), hunger would also be greater in the UPF group for meals offered with fixed energy. In addition, this large effect was also chosen because we had financial limitations, so a conservative estimate of the effect size would reduce the number of outcomes investigated and, consequently, the scope of our investigation. Nevertheless, considering the instrument used to assess hunger (scale), and the similarity of our interventions, we do believe now that our choice of “hunger” as the primary outcome was inadequate and that our study would only be able to detect differences in rate of intake, indeed. We recognize that this was an inconsistency in our planning, and we suggest that future research should include larger samples or use a cross-over design.
- While the authors attempted to match the meals for macronutrients and energy density, the classification of foods as UPF or non-UPF using the NOVA system may oversimplify the complexity of food processing. Recent literature has highlighted limitations of the NOVA classification system and suggested more nuanced approaches to categorizing processed foods (PMID: 35314769, PMID: 30820487, PMID: 28703086). The authors should discuss these limitations and consider using additional classification systems in future studies.
R- Thank you for your comment. Although we made a subtle point about the limitation of NOVA in section 4.2, we agree with the suggestion. We further added the following information: "Although the NOVA classification system is widely used in the literature and is adopted by some national guidelines, this system has known limitations in classifying the degree of food processing, mainly due to the lack of consistency in UPF definitions, and it is widely criticized [31,32]. We acknowledge that other classification systems could also be used, although there is still debate about their uses [33]." (Section 4.2, lines 442-447.
- The study focuses on acute effects of a single meal, which may not capture the long-term impacts of UPF consumption on health outcomes. The authors should acknowledge this limitation and discuss how their findings relate to studies examining chronic UPF intake, such as the work by Hall et al. (PMID: 31269427).
R- Thank you for the relevant comment. We added this point in the study limitations section (lines 567-572, "3) We recognize that our study results are based on an acute effect observed after a single meal. Therefore, we suggest caution when interpreting the data, as this specific assessment limits the generalization of the findings. Unlike studies that investigate the effects of chronic UPF consumption, ours does not allow us to capture potential long-term impacts of this consumption and its repercussions on health.”
- While the study measured various outcomes, it lacks a clear mechanistic framework to explain the observed differences between UPF and non-UPF meals. The authors should provide more in-depth discussion of potential mechanisms underlying their findings (PMID: 38294671, PMID: 38212644, PMID: 38418082), particularly regarding the faster intake rate and differences in appetitive measures as follows.
R- We appreciate your suggestion. As also suggested by the first reviewer, we further discussed our findings of intake rhythm and appetitive measures (section 4.3), including more references and potential mechanisms (texture of the food, familiarity, etc.) that explain such findings, as may be seen in the 1st, 2nd and 3rd paragraph of this section. Furthermore, we are grateful for the articles provided and, as requested, we have included your contributions in section 4.3 of the discussion (ref 38 and 39) and in the introduction (ref 3).
-Include additional measures of food reward and hedonic responses to provide a more comprehensive understanding of the appetitive effects of UPF vs. non-UPF meals.
R- Unfortunately, we do not have such information in this study. However, we appreciate the comment and consider it as a suggestion for future research.
-Explore the potential role of food texture and oral processing in mediating the observed differences in intake rate and appetitive responses.
R- Thank you for your comment. In our discussion, we have already briefly discussed the potential impact of different textures in the meals offered in our study on the final intake rate. Section 4.2, lines 437-439: and also in section 4.3. However, we have added new information to complement such issue: in Section 4.3, lines 471-476: “we also chose to match as much as possible the texture of our meals, based on previous findings by Teo et al. (2022) [36], who analyzed the impact of different textures of both UPF and minimally processed foods, and observed differences in the rate of intake and energy intake in healthy participants, suggesting that the softness characteristics of UPF could favor its faster rate of intake."
Furthermore, in Section 4.3, lines 480-487: “A systematic review with meta-analysis [37] indicated that a greater number of chews may be associated with reduced self-reported hunger and increased satiety. However, the physiological mechanisms that explain this relationship are not yet conclusive, although it is conjectured that the increase in the number of chews stimulates the release of gut hormones related to satiety. Although we did not directly analyze the relationship between these factors and found a significant effect only on eating capacity, our results may contribute to the understanding of how the level of food processing influences intake and the feeling of satiety. Although we did not directly analyze the relationship between these factors and found a significant effect only on eating capacity, our results may contribute to the understanding of how the level of food processing influences intake and the feeling of satiety."
-Conduct subgroup analyses to investigate potential sex differences in responses to UPF and non-UPF meals, as previous studies have suggested differential effects.
R- Due to our low statistical power, we opted not to conduct subgroup analysis. Still, we ran all analyses adjusting by sex, and all the results are essentially the same, with the exception of the log-transformed capacity to eat variable, which, in the mixed ANOVA adjusted by sex, ceases to show significant interaction between moment*group. We added this information in lines 346-347.
-Consider incorporating measures of gut hormones and microbiome composition in future studies to elucidate potential mechanisms underlying the metabolic effects of UPF consumption.
R—Thank you for your consideration. The present study already included measurements of intestinal hormones (ghrelin, GIP). However, another paper of our group addresses the analysis of intestinal microbiota.
- The study focuses on individuals with obesity, which may limit generalizability to other populations. Including a wider range of BMIs could provide more comprehensive insights.
R—Thanks for the comment. We inserted this information in the limitations (lines 572-574): "3) [...] Furthermore, the exclusive inclusion of individuals with obesity may represent an additional limiting factor in generalizing the results to other populations.".
Minor comments
- More detailed information on the specific foods used in each meal would be helpful for readers to better understand the intervention.
R- Although we appreciate the suggestion, we believe that the actual images of the meals offered in both groups (Figure 2a and 2b), together with the descriptions of the foods (Table 1), including the brands of the foods in the UPF group, can guide readers in understanding the interventions. Nevertheless, we added more details in section 2.5, lines 174-177: "The UPF meal consisted of commercial toast with cheese, ham, commercial strawberry jam, and margarine, as well as commercial guava juice with a fiber supplement. The control meal consisted of local bread toast with extra virgin olive oil, fried chicken eggs with soybean oil and salt, and guava fruit juice with honey."
- The manuscript mentions that meals were consumed between 8 and 10 am. Considering potential diurnal variations in metabolism, standardizing the meal time more strictly could improve consistency.
R: Thank you for your comment. We agree and will consider this standardization in future investigations.
- The exploratory analysis of leptin levels adjusted for sex is interesting but should be interpreted with caution due to the post-hoc nature of the analysis.
R—We appreciate and agree with your observation. Although the data is exploratory, more information has been added in the Results section (lines 384-391) and Discussion section (section 4.6, lines 540-553). We also highlighted that this is only a hypothesis-generating finding in our conclusions.
- Provide a more detailed rationale for the choice of outcomes measured and their relevance to understanding the health impacts of UPF.
R- We appreciate the concern raised, but we would like to reason with the reviewer that we have tried to address such an issue in our introduction section. We have shown the gaps in this area of investigation and indicated that such gaps need to be further explored to advance the scientific understanding of the relationship between UPF consumption and obesity. Our clinical trial, with carefully selected outcomes, seeks to provide more concrete answers than observational studies, which, although important, do not establish cause-and-effect relationships. The chosen outcomes cover key organ and metabolic systems linked to obesity, offering an integrated view of the impacts of acute UPF consumption. Below, we show the excerpts of the text that we believe are able to justify our choices and their impacts:
“Although the faster intake rate of UPF is shown in many studies, there is much heterogeneity in such findings, making this aspect an important target of investigation [8], given that it may trigger several physiological signs related to the perception of postprandial fullness and satiety, hindering the postprandial signaling to brain circuits and affecting different metabolic pathways [9,10]. Such pathways may include the secretion of satiety hormones. The structural changes in the UPF due to the levels of processing may facilitate the digestion process, possibly leading to altered hormonal responses and an imbalance in the levels of these hormones [11]“ (lines 64-71);
“[…] but some putative mechanisms that may underlie such an association. Postprandial energy expenditure (PPEE) may be one of these pathways [13,14]. Due to the lower complexity of the food matrix, UPF may require a shorter metabolization time and, therefore, yield a lower PPEE [12].” (lines 74-77);
“Furthermore, energy metabolism and body weight regulation are directly influenced by the autonomic nervous system (ANS), with evidence that obesity leads to a greater sympathetic tonus [15] (lines 80-82) […] which has been associated with obesity, eating behavior, and the metabolic impact of meals. However, the effect of UPF meals on HRV is unknown [16].” (lines 86-87).
- The manuscript would benefit from a more detailed description of the randomization process and allocation concealment. In addition, due to the nature of the intervention, the study could not be double-blinded. This potential source of bias should be more thoroughly addressed in the limitations section.
R- The randomization and allocation concealment procedures were better described in section 2.3. We also added the limitation of no blinding in the “4” limitation in the “limitations” section.
- The authors should provide more information on the validity and reliability of the methods used for measuring energy expenditure and autonomic function in this population.
R- We appreciate the comment and the opportunity to shed some light on the studies that used these methods and discussed their validity and reliability. Regarding autonomic function, we include here a compilation of studies that involved different audiences and used the same device used in our study (PMID: 36081005, PMID: 32163924, PMID: 35172251 and PMID: 24136753), showing that the literature already addresses the use of these methods. Likewise, we include studies on energy expenditure (PMID: 16869134, PMID: 24051107 and PMID: 23541105).
12 The discussion section could be strengthened by comparing the findings more extensively with existing literature on UPF consumption and its effects on appetite and metabolism.
R- Thanks for the suggestions. As mentioned in comment 4, we have improved the discussion on these aspects.
- The authors should address potential confounding factors, such as habitual diet, physical activity levels, and sleep patterns, which could influence the measured outcomes.
R- We appreciate your comment. Although we understand that confounding factors may play a role, we believe that the fact that our study is a randomized controlled trial, confounding becomes less of a concern given that, by definition, a confounder must interfere both with the exposure and with the outcome and in the case of an RCT, the exposure (in our case, the meal) is, by definition, determined by randomness. Hence, all backdoor paths are closed, and there is no true confounding. Therefore, we opted not to conduct multivariable analysis.
- Provide a more detailed rationale for the choice of outcomes measured and their relevance to understanding the health impacts of UPF.
R- We clarify this in point 9 and thank you once again.
- Consider including a table or figure summarizing the main findings to improve the clarity of result presentation.
R- Thank you for your suggestion. We will include it as a graphical abstract.
- Discuss the implications of the findings for public health and dietary recommendations, while acknowledging the limitations of the acute, single-meal design.
R—We appreciate this suggestion. As the reviewer pointed out, our study is of an acute single-meal intervention that is not able to capture long-term effects. We are not confident enough in our study design to infer implications for public health and dietary recommendations. Therefore, we opt to stick with metabolic discussion, considering such an acute intervention. As pointed out in the response to commentary number 3, we have added such information in the limitations section.
- Expand the introduction or discussion section by citing the following relevant articles (PMID: 36517001, PMID: 32674529).
R - We appreciate the suggestion. As both papers suggested by the reviewer deal with nutrition and NAFLD, we opted not to include such papers in our introduction/discussion, given that we believe they do not fulfill our scope.
Round 2
Reviewer 2 Report
Comments and Suggestions for Authors
Nothing to add.